# LOW LATENCY PRIVACY PRESERVING INFERENCE

## ABSTRACT

When applying machine learning to sensitive data one has to balance between accuracy, information leakage, and computational-complexity. Recent studies have shown that Homomorphic Encryption (HE) can be used for protecting against information leakage while applying neural networks. However, this comes with the cost of limiting the kind of neural networks that can be used (and hence the accuracy) and with latency of the order of several minutes even for relatively simple networks. In this study we improve on previous results both in the kind of networks that can be applied and in terms of the latency. Most of the improvement is achieved by novel ways to represent the data to make better use of the capabilities of the encryption scheme.

## 1    INTRODUCTION

Machine learning is used in domains such as education, health, and finance in which data may be private or confidential. Therefore, machine learning algorithms should preserve privacy while making accurate predictions. The privacy requirement pertains to all sub-tasks of the learning process, such as training and inference. In this work, we focus on private neural-networks inference. In this problem, popularized by the work on CryptoNets (Dowlin et al., 2016), the goal is to build an inference service that can make predictions on private data. To achieve this goal, the data is encrypted before it is sent to the prediction service which should be capable of operating on the encrypted data without having access to the raw content. To allow that, several cryptology technologies have been proposed, including Secure Multi-Party Computation (MPC) (Yao, 1982; Goldreich et al., 1987), hardware enclaves, such as Intel's Software Guard Extensions (SGX) (McKeen et al., 2013), Homomorphic Encryption (Gentry, 2009), and combinations of these techniques.

The different approaches present different trade-offs in terms of computation, accuracy, and security. HE presents the most stringent security model. The security assumption relies on the hardness of solving a mathematical problem for which there are no known efficient algorithms, even in the presence of quantum computers (Gentry, 2009; Albrecht et al., 2018). Other techniques, such as MPC and SGX make additional assumptions and therefore provide a weaker sense of protection to the data (Yao, 1982; McKeen et al., 2013; Chen et al., 2018; Koruyeh et al., 2018).

While HE provides the highest level of security it is also limited in the kind of operations it allows and the complexity of these operations (see Section 1.1). CryptoNets (Dowlin et al., 2016) was the first demonstration that it may be feasible to use HE to build privacy preserving Encrypted Prediction as a Service (EPaaS) solutions (Sanyal et al., 2018). CryptoNets are capable of making predictions with accuracy of $99\%$ on the MNIST task (LeCun et al., 2010) such that each prediction takes 250 seconds to complete. CryptoNets are also capable of packing 4096 prediction requests and operate on all of them in parallel which allows throughput of $\sim 59000$ predictions per hour.

CryptoNets have several limitations that we address in this work, the first of them is latency. CryptoNets provide high throughput by operating on 4096 instances in parallel, however, all these instances have to come from a single source and use the same secret key. Therefore, this capability may be of little use in practice. Thus, we trade the high throughput in favor of low latency and show that the same neural network that was used by CryptoNets can be evaluated in as little as 2.2 seconds. We show that a part of this gain is an "engineering gain" which is a result of using a more recent implementation of HE. However, this "engineering gain" accounts for only $10\times$ speedup. Most of the speedup comes from a new way to represent data when applying neural-networks using HE which we call LoLa. In a nut-shell, CryptoNets represent each node in the neural network as a separate

message for encryption, while LoLa encrypts entire layers which results in a $11.2\times$ speedup on top of the "engineering gain". Together, these improvements results in a $114\times$ improvement in latency while maintaining the same level of security and accuracy.[1]

LoLa provides another significant benefit over CryptoNets. Since CryptoNets encode every node in the network as a separate message, they create a memory bottleneck when applied to networks with many nodes. We demonstrate that in an experiment conducted on the CIFAR-10 dataset for which the CryptoNets approach fails to execute since it requires 100's of Gigabytes of RAM. However, the low-latency approach, LoLa, which encodes layers instead of nodes, can make predictions in 12 minutes using only few Gigabytes of RAM.

The experiment on CIFAR demonstrates that the LoLa approach can handle larger networks than CryptoNets. However, there is still a big penalty for the size of the network: predictions on MNIST are achieved in 2.2 seconds, and this latency jumps to 12 minutes for the slightly more complex task in the CIFAR-10 dataset. Therefore, it is reasonable to ask whether any of these approaches can scale to handle tasks such as analyzing large and complex images. To that extent, we propose another solution which represents the input to the network using semantically meaningful features instead of pixels. These semantically meaningful features are extracted using the convolution layers of standard networks such as AlexNet (Krizhevsky et al., 2012). We consider these networks as "standard libraries" for machine learning tasks. Using such features allows reducing the size of the message to be sent and the complexity of the network that is needed for classification. Indeed, we use this approach to demonstrate private predictions in 0.18 seconds on the CalTech-101 dataset with class balanced accuracy of $75.7\%$.

## 1.1 Homomorphic Encryption

In this work we use Homomorphic Encryptions (HE) to provide privacy (we refer the reader to Dowlin et al. (2017) for a more comprehensive introduction). HEs are encryptions that allow operating on data while it is encrypted without requiring access to the secret key (Gentry, 2009). The data used for encryption is assumed to be elements in a ring $\mathcal{R}$. On top of the encryption function $\mathbb{E}$ and the decryption function $\mathbb{D}$, the HE scheme provides two additional operators $\oplus$ and $\otimes$ such that for any $x_1, x_2 \in \mathcal{R}$

$$\mathbb{D}\left(\mathbb{E}(x_1) \oplus \mathbb{E}(x_2)\right) = x_1 + x_2 \text{ and}$$
$$\mathbb{D}\left(\mathbb{E}(x_1) \otimes \mathbb{E}(x_2)\right) = x_1 \times x_2$$

where $+$ and $\times$ are the standard addition and multiplication operations on the ring $\mathcal{R}$. Therefore, the $\oplus$ and $\otimes$ operators allow computing addition and multiplication operators on the data in its encrypted form and thus computing any polynomial function.

Since Gentry's seminal paper, in which he introduced the first HE scheme (Gentry, 2009), additional schemes have been proposed. In this work we use the Brakerski/Fan-Vercauteren scheme (BFV) (Fan & Vercauteren, 2012; Brakerski & Vaikuntanathan, 2014) as it is implemented in the SEAL library version 2.3.1.[2] In this scheme, the ring on which the Homomorphic Encryption operates is $\mathcal{R} = \frac{\mathbb{Z}_p[x]}{x^n+1}$ where $\mathbb{Z}_p = \frac{\mathbb{Z}}{p\mathbb{Z}}$. If the parameters $p$ and $n$ are chosen such that there is an order $2n$ root of unity in $\mathbb{Z}_p$, then every element in $\mathcal{R}$ can be viewed as a vector of dimension $n$ of elements in $\mathbb{Z}_p$ where addition and multiplication operate component-wise (Brakerski et al., 2014). In this view, the BFV scheme allows a another operation on the encrypted data: rotation. The ideal rotation operation of size $k$ sends the value in the $i$'th coordinate of a vector to the $((i + k) \mod n)$ coordinate. The BFV scheme allows a slight modified version of the ideal rotation (see Appendix A) but for the sake of our discussion this detail is insignificant.

## 1.2 Related Work

The task of private predictions gained significant attention in recent years. Dowlin et al. (2016) presented CryptoNets which demonstrated the feasibility of private neural networks predictions using

---

[1]The HE scheme used by CryptoNets was found to have some weaknesses that the HE scheme that we use does not suffer from.

[2]The SEAL library can be found at `http://sealcrypto.org/`. We use parameters that guarantee 128 bits of security according to the proposed standard for Homomorphic Encryptions (Albrecht et al., 2018).

HE. CryptoNets are capable of making predictions with high throughput but are limited in both the depth of the network they can support and the latency per prediction. Bourse et al. (2017) used a different HE scheme that allows fast bootstrapping which results in only linear penalty for additional layers in the network. However, it is slower per operation and therefore, the results they presented on the MNIST data-set use small models with significantly lower accuracy (see Table 1). Sanyal et al. (2018) argued that many of these methods leak information about the structure of the neural-network that the service provider uses through the parameters of the encryption. They presented a method that leaks less information about the neural-network but their solution is orders of magnitude slower. Nevertheless, their solution has the nice benefit that it allows the service provider to change the network without requiring changes in the client side.

Other researchers proposed using different encryption schemes. For example, the Chameleon system (Riazi et al., 2018) uses MPC to demonstrate private predictions on MNIST and Juvekar et al. (2018) use a hybrid MPC-HE approach for the same task. Hardware based solutions were also proposed, for example, Tramer & Boneh (2018). Some of these approaches provide faster predictions which are, in some cases, more accurate, however, this comes with the cost of a using a lower level of security.

## 2 DATA REPRESENTATION

Feed-forward neural networks are functions that can be computed by an alternating sequence of linear transformations and non-linear transformations. Linear transformations include dense, convolution layers and average pooling layers. Non-linear transformations include activation functions and max pooling layers. In most cases, we can consider this sequence to be alternating between linear transformations and non-linear ones since consecutive linear transformations can be combined into a single linear transformation and sequences of non-linear transformations can be merged as well.

For most of this work, we restrict the non-linear transformations to the square activation function. This follows CryptoNets (Dowlin et al., 2016) that showed that high accuracy can be achieved even with this restriction. We demonstrate this again on the CIFAR data-set in Section 4. Recall that HE supports point-wise multiplication of vectors and therefore it is straight-forward to implement the square activation function.

The main linear transformations we consider are dot-products and matrix-vector multiplications. Given two vectors, we can implement a dot product between two vectors whose size is a power of 2 by first applying point-wise multiplication between the two vectors and then a series of $\log n$ rotations of size $1, 2, 4, \ldots, n/2$ and addition between each rotation. The result of such a dot product operation is a vector that holds the results of the dot-product in all its coordinates.[3]

The dot-product operation can induce a change in representations. For example, given a weights matrix and an input vector represented as a single message, we can multiply the matrix by the vector using $r$ dot-product operations where $r$ is the number of rows in the matrix. The result of this operation is a vector of length $r$ that is spread across $r$ messages. Therefore, the result has a different representation than the representation of the input vector. Different representations can induce different computational costs and therefore choosing the right representations throughout the computation is important for computational efficiency. It is possible to change representations but this requires additional computational steps. Instead, we propose using various representations in the network inference. We start our discussion by presenting different possible vector representations.

### 2.1 VECTOR REPRESENTATIONS

Recall that a message in HE can be thought of as a vector of length $n$ of elements in $\mathbb{Z}_p$. For the sake of brevity, we assume that the dimension of the vector $\mathbf{v}$ to be encoded is of length $k$ such that $k \leq n$, for otherwise multiple messages can be combined. For any vector $\mathbf{u}$ we denote by $u_i$ its $i^{\text{th}}$ coordinate.

---

[3]For example, consider calculating the dot product of 4-dimensional vectors $(v_1, ..., v_4)$ and $(w_1, ..., w_4)$. Point-wise multiplication, rotation of size 1 and summation results in the vector $(v_1 w_1 + v_4 w_4, v_2 w_2 + v_1 w_1, v_3 w_3 + v_2 w_2, v_4 w_4 + v_3 w_3)$. Another rotation of size 2 and summation results in the 4 dimensional vector which contains the dot-product of the vectors in all coordinates.

**2.1.1 Dense representation:** A vector $\mathbf{v}$ is represented as a single message $\mathbf{m}$ by setting $v_i \mapsto m_i$.

**2.1.2 Sparse representation:** A vector $\mathbf{v}$ of length $k$ is represented in $k$ messages $\mathbf{m}^1, \ldots \mathbf{m}^k$ such that $\mathbf{m}^i$ is a vector in which every coordinate is set to $v_i$.[4]

**2.1.3 Stacked representation:** For a short (low dimension) vector $\mathbf{v}$, the stacked representation holds several copies of the vector $\mathbf{v}$ in a single message $\mathbf{m}$. Typically this will be done by finding $d = \lceil \log(k) \rceil$, the smallest $d$ such that the dimension of $\mathbf{v}$ is at most $2^d$ and setting $m_i, m_{i+2^d}, m_{i+2 \cdot 2^d}, \ldots = v_i$.

**2.1.4 Interleaved representation:** The interleaved representation uses a permutation $\sigma$ of $[1, \ldots, n]$ to set $m_{\sigma(i)} = v_i$. The dense representation can be viewed as a special case of the interleaved representation where $\sigma$ is the identity permutation.

**2.1.5 Convolution representation:** This is a special representation that makes convolution operations efficient. A convolution, when flattened to a single dimension, can be viewed as a restricted linear operation where there is a weight vector $\mathbf{w}$ of length $r$ (the window size) and a set of permutations $\sigma_i$ such that the $i$'th output of the linear transformation is $\sum_j w_j v_{\sigma_i(j)}$. The convolution representation takes a vector $v$ and represents it as $r$ messages $\mathbf{m}^1, \ldots, \mathbf{m}^r$ such that $m_i^j = v_{\sigma_i(j)}$.[5]

**2.1.6 SIMD representation:** CryptoNets (Dowlin et al., 2016) represent each data element as a separate message but maps multiple data vectors into the same set of messages. More details about this representation are in Appendix B.

## 2.2 MATRIX-VECTOR MULTIPLICATIONS

Matrix-vector multiplication is a core operation in neural networks. The matrix may contain the learned weights of the network and the vector represents the values of the nodes at a certain layer. Here we present different ways to implement such matrix-vector operations. Each method operates on vectors in different representations and produces output in yet another representation. Furthermore, the weight matrix has to be represented appropriately as a set of vectors, either column-major or row-major to allow the operation. We assume that the matrix $W$ has $k$ columns $\mathbf{c}^1, \ldots, \mathbf{c}^k$ and $r$ rows $\mathbf{r}^1, \ldots, \mathbf{r}^r$.

**2.2.1 Dense Vector – Row Major:** If the vector is given as a dense vector and each row $\mathbf{r}^j$ of the weight matrix is encoded as a dense vector then the matrix-vector multiplication can be applied using $r$ dot-product operations. As already described above, a dot-product requires a single multiplication and $\log(n)$ additions and rotations. The result is a sparse vector of length $r$.

**2.2.2 Sparse Vector – Column Major:** Recall that $W\mathbf{v} = \sum v_i \mathbf{c}^i$. Therefore, when $\mathbf{v}$ is encoded in a sparse format, the message $\mathbf{m}^i$ has all its coordinate set to $v_i$ and $v_i \mathbf{c}^i$ can be computed using a single point-wise multiplication. Therefore, $W\mathbf{v}$ can be computed using $k$ multiplications and additions and the result is a dense vector.

**2.2.3 Stacked Vector – Row Major:** For the sake of clarity, assume that $k = 2^d$ for some $d$. In this case $n/k$ copies of $\mathbf{v}$ can be stacked in a single message $\mathbf{m}$ (this operation requires $\log(n/k) - 1$ rotations and additions). By concatenating $n/k$ rows of $W$ into a single message a special version

---

[4]Recall that the encrypted messages are in the ring $R = \frac{\mathbb{Z}_p[x]}{x^n+1}$ which, by the choice of parameters, is homomorphic to $(\mathbb{Z}_p)^n$. When a vector has the same value $v_i$ in all its coordinates, then its polynomial representation in $\frac{\mathbb{Z}_p[x]}{x^n+1}$ is the constant polynomial $v_i$.

[5]For example, consider a matrix $A \in \mathbb{R}^{4 \times 4}$ which corresponds to an input image and a $2 \times 2$ convolution filter that slides across the image with stride 2 in each direction. Let $a_{i,j}$ be the entry at row $i$ and column $j$ of the matrix $A$. Then, in this case $r = 4$ and the following messages are formed $M^1 = (a_{1,1}, a_{1,3}, a_{3,1}, a_{3,3})$, $M^2 = (a_{1,2}, a_{1,4}, a_{3,2}, a_{3,4})$, $M^3 = (a_{2,1}, a_{2,3}, a_{4,1}, a_{4,3})$ and $M^4 = (a_{2,2}, a_{2,4}, a_{4,2}, a_{4,4})$. In some cases it will be more convenient to combine the interleaved representation with the convolution representation by a permutation $\tau$ such that $m_{\tau(i)}^j = v_{\sigma_i(j)}$.

of the dot-product operation can be used to compute $n/k$ elements of $W\mathbf{v}$ at once. First, a point-wise multiplication of the stacked vector and the concatenated rows is applied followed by $d-1$ rotations and additions where the rotations are of size $1, 2, \ldots, 2^{d-1}$. The result is in the interleaved representation.[6]

The Stacked Vector - Row Major gets its efficiency from two places. First, the number of modified dot product operations is $rk/n$ and each dot product operation requires a single multiplication and second, only $d$ rotations and additions (compared to $\log n$ rotations and additions in the standard dot-product procedure).

**2.2.4 Interleaved Vector – Row Major:** This setting is very similar to the dense vector – row major matrix multiplication procedure with the only difference being that the columns of the matrix have to be shuffled to match the permutation of the interleaved representation of the vector. The result is in sparse format.

**2.2.5 Convolution vector – Row Major:** A convolution layer applies the same linear transformation to different locations on the data vector $\mathbf{v}$. For the sake of brevity, assume the transformation is one-dimensional. In neural network language that would mean that the kernel has a single map. Obviously, if more maps exist, then the process described here can be repeated multiple times.

Recall that a convolution, when flattened to a single dimension, is a restricted linear operation where the weight vector $\mathbf{w}$ is of length $r$, and there exists a set of permutations $\sigma_i$ such that the $i$'th output of the linear transformation is $\sum w_j v_{\sigma_i(j)}$. In this case, the convolution representation is made of $r$ messages such that the $i$'th element in the message $\mathbf{m}^j$ is $v_{\sigma_i(j)}$. By using a sparse representation of the vector $\mathbf{w}$, we get that $\sum w_j \mathbf{m}^j$ computes the set of required outputs using $r$ multiplications and additions. When the weights are not encrypted, the multiplications used here are relatively cheap since the weights are scalar and BFV supports fast implementation of multiplying a message by a scalar. The result of this operation is in a dense format.

# 3 SECURE NETWORKS FOR MNIST

The neural network used for the MNIST data-set (LeCun et al., 2010) is the same network used by CryptoNets (Dowlin et al., 2016). After suppressing adjacent linear layers it can be presented as a $5 \times 5$ convolution layer with a stride of $(2, 2)$ and 5 output maps, which is followed by a square activation function that feeds a fully connected layer with 100 output maps, another square activation and another fully connected layer with 10 outputs (see Figure 2 in the appendix).

The baseline implementation uses the techniques presented in CryptoNets in Table 1. Recall that CryptoNets use the SIMD representation (Section 2.1.6) in which each pixel requires its own message. Therefore, since each image in the MNIST data-set is made of an array of $28 \times 28$ pixels, the input to the CryptoNets network is made of 784 messages. On the reference machine used for this work (Azure standard B8ms virtual machine with 8 vCPUs and 32GB of ram) the original CryptoNets implementation runs in 205 seconds. Re-implementing it to use better memory management and multi-threading in SEAL 2.3 reduces the running time to 24.8 seconds. Since this implementation allows batching of 8192 images to be processed simultaneously, it has a potential throughput of 1189161 predictions per hour which is, as far as we know, the highest throughput reported on this task by a large margin.

While CryptoNets provide high throughput, in many cases, it is hard to utilize this high throughput which requires batching together 8192 requests from sources that share the same secret key. If each user has only a single record to be predicted on, the throughput is governed by the latency and therefore, we move towards reducing latency. We do that by replacing the SIMD representation with other representations. As a result, throughput is sacrificed in favor of latency.

The Low-Latency CryptoNets (LoLa) uses the same network layout and has accuracy of $98.95\%$ (see Table 2 for a summary of the data representations used by LoLa). However, it is implemented

---

[6]For example, consider a $2 \times 2$ matrix $W$ flattened to a vector $\mathbf{w} = (w_{1,1}, w_{1,2}, w_{2,1}, w_{2,2})$ and a two-dimensional vector $\mathbf{v} = (v_1, v_2)$. Then, after stacking the vectors, point-wise multiplication, rotation of size 1 and summation, the second entry of the result contains $w_{1,1}v_1 + w_{1,2}v_2$ and the fourth entry contains $w_{2,1}v_1 + w_{2,2}v_2$. Hence, the result is in an interleaved representation.

| Method | Accuracy | Latency | Throughput | |
|---|---|---|---|---|
| FHE–DiNN100 | 96.35% | 1.65 | 2182 | (Bourse et al., 2017) |
| LoLa-Small | 96.92% | **0.29** | 12500 | |
| CryptoNets | 98.95% | 250 | 58982 | (Dowlin et al., 2016) |
| CryptoNets 2.3 | 98.95% | 24.8 | **1189160** | |
| LoLa | 98.95% | 7.2 | 500 | |
| LoLa-Conv | 98.95% | **2.2** | 1636 | |

Table 1: MNIST performance comparison. Solutions are grouped by accuracy levels.

differently: the input to the network is a single dense message where the pixel values are mapped to coordinates in the encoded vector line after line . The first step in processing this message is breaking it into 25 messages corresponding to the 25 pixels in the convolution map to generate a convolution representation. Creating each message requires a single vector multiplication. This is performed by creating 25 masks. The first mask is a vector of zeros and ones that corresponds to a matrix of size $28 \times 28$ such that a one is in the $(i, j)$ coordinate if the $i, j$ pixel in the image appears as the upper left corner of the $5 \times 5$ window of the convolution layer. Multiplying point-wise the input vector by the mask creates the first message in the convolution representation as described in Section 2.1.5 hybrided with the interleaved representation as described in footnote 5. Similarly the other messages in the convolution representation are created. Note that all masks are shifts of each other which allows using the convolution representation-row major multiplication to implement the convolution layer (see Section 2.2.5). To do that, think of the 25 messages as a matrix and the weights of a map of the convolution layer as a sparse vector. Therefore, the outputs of the entire map can be computed using 25 multiplications (of each weight by the corresponding vector) and 24 additions. Note that there are 169 windows and all of them are computed simultaneously. However, the process repeats 5 times for the 5 maps of the convolution layer.

The result of the convolution layer are 5 messages, each one of them contains 169 results. They are united into a single vector by rotating the messages such that they will not have active values in the same locations and summing the results. At this point, a single message holds all the 845 values (169 windows $\times 5$ maps). This vector is squared, using a single multiplication operation, to implement the activation function that follows the convolution layer. This demonstrates one of the main differences between CryptoNets and LoLa; In CryptoNets, the activation layer requires 845 multiplication operations, whereas in LoLa it is a single multiplication. Even if we add the manipulation of the vector to place all values in a single message, as described above, we add only 4 rotations and 4 additions which are still much fewer operations than in CryptoNets.

Next, we apply a dense layer with 100 maps. LoLa uses messages of size $n = 16384$ where the 845 results of the previous layer, even though they are in interleaving representation, take fewer than 1024 dimensions. Therefore, 16 copies are stacked together which allows the use of the Stacked vector – Row Major multiplication method. This allows computing 16 out of the 100 maps in each operation and therefore, the entire dense layer is computed in 7 iterations resulting in 7 interleaved messages. By shifting the $i^{th}$ message by $i - 1$ positions, the active outputs in each of the messages are no longer in the same position and they are added together to form a single interleaved message that contains the 100 outputs. The following square activation requires a single point-wise-multiplication of this message. The final dense layer is applied using the Interleaved vector – Row Major method to generate 10 messages, each of which contains one of the 10 outputs.[7]

Overall, applying the entire network takes only 7.2 seconds on the same reference hardware which is 34.7× faster than CryptoNets and 3.4× faster than CryptoNets 2.3. This result can be further improved by changing the input to the network; Instead of taking as an input a dense representation of the image, the LoLa-Conv network takes as its input 25 messages which are the convolution representation of the image. This removes a processing step which saves time but also reduces the

---

[7]It is possible, if required, to combine them into a single message in order to save communication.

amount of noise accumulated during the computation and allows working with messages of size $n = 8192$, which further reduces the computation time.

The LoLa-Conv starts with a convolution vector – row major multiplication for each of the 5 maps of the convolution layer. The 5 dense output messages are joined together with a rotation and addition to form a single dense vector of 845 elements. This vector is squared using a single multiplication and 8 copies of the results are stacked before applying the dense layer as 13 rounds of Stacked vector – Row Major multiplication. The 13 vectors of interleaved results are rotated and added to form a single interleaved vector of results which is squared using a single multiplication. Finally, Interleaved vector – Row Major multiplication is used to obtain the final result. This version computes the entire network in only 2.2 seconds which is $3.3\times$ faster than LoLa, $11\times$ faster than CryptoNets 2.3 and $114\times$ faster than CryptoNets. See Table 3 for a summary of the data representations used by LoLa-Conv.

Table 1 shows a summary of the performance of different methods and more details can be found in Appendix C. Bourse et al. (2017) showed faster results with similar security level, albeit with lower accuracy. To compare with that, LoLa-Small is similar to Lola-Conv but has only a convolution layer, square activation and a dense layer. This solution is more accurate than the networks used by Bourse et al. (2017) and at the same time it is $5.5\times$ faster.

## 4 SECURE NETWORKS FOR CIFAR

The Cifar-10 data-set (Krizhevsky & Hinton, 2009) presents a more challenging task of recognizing one of 10 different types of objects in a small image. The neural network used has the following layout: the input is a $3 \times 32 \times 32$ image (i) $3 \times 3$ linear convolution with stride of $(1, 1)$ and 128 output maps, (ii) $2 \times 2$ average pooling with $(2, 2)$ stride (iii) $3 \times 3$ convolution with $(1, 1)$ stride and 83 maps (iv) Square activation (v) $2 \times 2$ average pooling with $(2, 2)$ stride (vi) $3 \times 3$ convolution with $(1, 1)$ stride and 163 maps (vii) Square activation (vii) $2 \times 2$ average pooling with stride $(2, 2)$ (viii) fully connected layer with 1024 outputs (ix) fully connected layer with 10 outputs (x) softmax. ADAM was used for optimization (Kingma & Ba, 2014) together with dropouts after layers (vii) and (viii). We use zero-padding in layers (i) and (vii). See Figure 3 for an illustration of the network.

For inference, adjacent linear layers were collapsed to form the following structure: (i) $8 \times 8 \times 3$ convolutions with a stride of $(2, 2, 0)$ and 83 maps (ii) square activation (iii) $6 \times 6 \times 83$ convolution with stride $(2, 2, 0)$ and 163 maps (iv) square activation (v) dense layer with 10 output maps. This network is much larger than the network used for MNIST by CryptoNets. The input to the CIFAR network has 3072 nodes, the first hidden layer has 16268 nodes and the second hidden layer has 4075 nodes (compared to 784, 845, and 100 nodes respectively for MNIST).[8] The accuracy of this network is 74.1% and it uses plain-text modulus $p = 2148728833 \times 2148794369 \times 2149810177$ (the factors are combined using the Chinese Reminder Theorem) and $n = 16384$. See Figure 4 for an illustration of this network.

Due to the sizes of the hidden layers, implementing this network with SIMD representation requires more memory than available on the reference machine, since the SIMD representation requires a message for each node in each layer. Therefore, we used the LoLa-Conv approach to implement this network. The image is encoded using the convolution representation into $3 \times 8 \times 8 = 192$ messages. The convolution layer is implemented using the convolution vector – row major matrix-vector multiplication technique. The results are combined into a single message using rotations and additions which allows the square activation to be performed with a single point-wise multiplication. The second convolution layer is performed using row major-dense vector multiplication. Although this layer is a convolution layer, each window of the convolution is so large that it is more efficient to implement it as a dense layer. The output is a sparse vector which is converted into a dense vector by point-wise multiplications and additions which allows the second square activation to be performed with a single point-wise multiplication. The last dense layer is implemented with a row major-dense vector technique again resulting in a sparse output.

Executing this network takes 730 seconds out of which the second layer consumes 711 seconds. Therefore, for this task the bottleneck in performance is the sizes of the weight matrices and data

---

[8]The map counts of the different layers were selected such that the sizes of the hidden layers will fit inside the encrypted messages.

vectors as evident by the number of parameters which is $< 90,000$ in the MNIST network and $> 500,000$ in the CIFAR network. In the following section we present an approach to mitigate this problem.

## 5 Applying Deep Nets using Deep Representations

Homomorphic Encryption has two main limitations when used for evaluating deep networks: noise growth and message size growth. Noise growth is a result of the number of operations that has to take place. Each such operation increases the noise in the encrypted message and when this noise becomes too large, it is no longer possible to decrypt the message correctly. This problem can be mitigated using bootstrapping, while taking a performance hit. The message size grows with the size of the network as well. Since, in its core, the HE scheme operates in $\mathbb{Z}_p$, the parameter $p$ has to be selected such that the largest number obtained during computation would be smaller than $p$. Since every multiplication might double the required size of $p$, it has to grow exponentially with respect to the number of layers in the network. The recently introduced HEAAN scheme (Cheon et al., 2017) is more tolerant towards message growth but even HEAAN would not be able to operate efficiently on very deep networks.

We propose solving both the message growth and the noise growth problems using deep representations. Instead of encrypting the data in its raw format, it is first converted, by a standard network, to create a deep representation. For example, if the data is an image, then instead of encrypting the image as an array of pixels, a network, such as AlexNet (Krizhevsky et al., 2012), VGG (Simonyan & Zisserman, 2014), or ResNet (He et al., 2016), first extracts a deep representation of the image, using one of its last layers. The resulting representation is encrypted and sent for evaluation. This approach has several advantages. First, this representation is small even if the original image is large. Moreover, with deep representations it is possible to obtain high accuracies using shallow networks: in most cases a linear predictor is sufficient which translates to a fast evaluation with HE. It is also a very natural thing to do since in many cases of interest, such as in medical image, training a very deep network from scratch is almost impossible since data is scarce. Hence, it is a common practice to use deep representation and train only the top layer(s) (Yosinski et al., 2014).

To test the deep representation approach we used AlexNet (Krizhevsky et al., 2012) to generate features and trained a linear model to make predictions on the CalTech-101 data-set (Fei-Fei et al., 2006).[9] See Table 4 for a summary of the data representations used for the CalTech-101 dataset. Since the CalTech-101 dataset is not class balanced, we used only the first 30 images from each class where the first 20 where used for training and the other 10 examples where used for testing. The obtained model has class-balanced accuracy of $75.7\%$. The inference time, on the encrypted data, takes only $0.178$ seconds when using the dense vector – row major multiplication.

## 6 Conclusions

The problem of privacy in machine learning is gaining importance due to legal requirements and greater awareness to the benefits and risks of machine learning systems. The task of private inference, specifically with neural networks, serves as a benchmark and catalyst to promote further study in this domain. In this work, we showed how data representations can be used to accelerate private predictions using Homomorphic Encryption. We demonstrated both the ability to operate on more complex networks as well as lower latency on networks that were already studied in the past.

Some of the methods we propose require precomputation on the client side. In many cases, HE is presented as a method to offload computation from a power-limited client to the cloud. However, this is not the only reason to use privacy preserving prediction services: in some applications the data is sensitive while the service provider is not willing to share the model which may be a result of a costly development process. In these cases, the techniques we present here allow the provider to offer its services while respecting the privacy of data.

---

[9]More complex classifiers did not improve accuracy.

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

## A ROTATIONS

For the rotation operation in the BFV encryption scheme it is easier to think of the message as a $2 \times n/2$ matrix:

$$\begin{bmatrix} m_1 & m_2 & \cdot & \cdot & m_{n/2} \\ m_{n/2+1} & m_{n/2+2} & \cdot & \cdot & m_n \end{bmatrix}$$

with this representation in mind, there are two rotations allowed, one switches the row, which will turn the above matrix to

$$\begin{bmatrix} m_{n/2+1} & m_{n/2+2} & \cdot & \cdot & m_n \\ m_1 & m_2 & \cdot & \cdot & m_{n/2} \end{bmatrix}$$

and the other rotates the columns. For example, rotating the original matrix by one column to the right will result in

$$\begin{bmatrix} m_{n/2} & m_1 & \cdot & \cdot & m_{n/2-1} \\ m_n & m_{n/2+1} & \cdot & \cdot & m_{n-1} \end{bmatrix} \ .$$

Since $n$ is a power of two, and the rotations we are interested in are powers of two as well, for the sake of this work, thinking about the rotations as simple rotations of the elements in the message yields similar results. In this view, the row-rotation is a rotation of size $n/2$ and smaller rotations are achieved by column rotations.

## B THE SIMD REPRESENTATION

The vector structure of messages used by CryptoNets allow parallel execution over multiple data simultaneously. CryptoNets takes $n$ input vectors $\mathbf{v}^1, \ldots, \mathbf{v}^n$ and creates a dense representation in which these $n$ messages of length $k$ are encoded in $k$ messages $\mathbf{m}^1, \ldots, \mathbf{m}^k$ such that $m_i^j = v_j^i$. All operations between vectors and matrices are implemented using additions and multiplications only. For example, a dot product between two vectors of length $k$ is implemented by $k$ multiplications and additions. Therefore, it acts as a sparse representation.

The advantage of this representation, which we call the *SIMD Representation,* is that the cost of applying an operation to a vector is the same cost of applying the same operation to $n$ vectors, hence

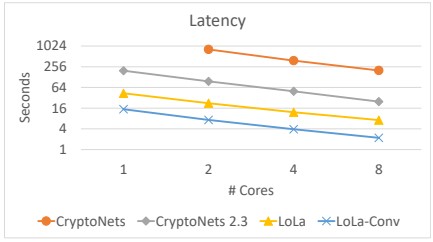 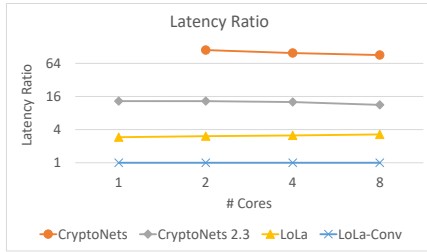

Figure 1: The latency of the different network implementations for the MNIST task with respect to the number of available cores. The right figure shows the ratio between the latency of each solution and the latency of the LoLa-Conv

it supports the Single Instruction Multiple Data (SIMD) framework. However, it is costly in two ways: the computational complexity of multiplying a matrix of size $r \times k$ with a vector of length $k$ is $O\left(rk\right)$ HE operations, and the memory consumption is large as well since a vector of length $k$ requires $k$ messages. In this sense it is similar to the sparse representation. However, the ability to perform SIMD operations provides it with high throughput, much like the dense representation.

## C    PARALLEL SCALING

The performance of the different solutions is affected by the amount of parallelism allowed. The hardware used for experimentation in this work has 8 cores. Therefore, we tested the performance of the different solutions with 1, 2, 4, and 8 cores to see how the performance varies. The results of these experiments are presented in Figure 1. These results show that at least up to 8 cores the performance of all methods scales linearly when tested on the MNIST data-set. This suggests that the latency can be further improved by using machines with higher core count.

## D    LOLA REPRESENTATION CHANGES

The following tables show the different stages that the data goes through when using LoLa. This illustrates how the different representations are used during the computation. Table 2 shows the process that LoLa applies, Table 3 shows the process for LoLa-Conv, and Table 4 shows the process for the method proposed for processing the CalTech-101 dataset.

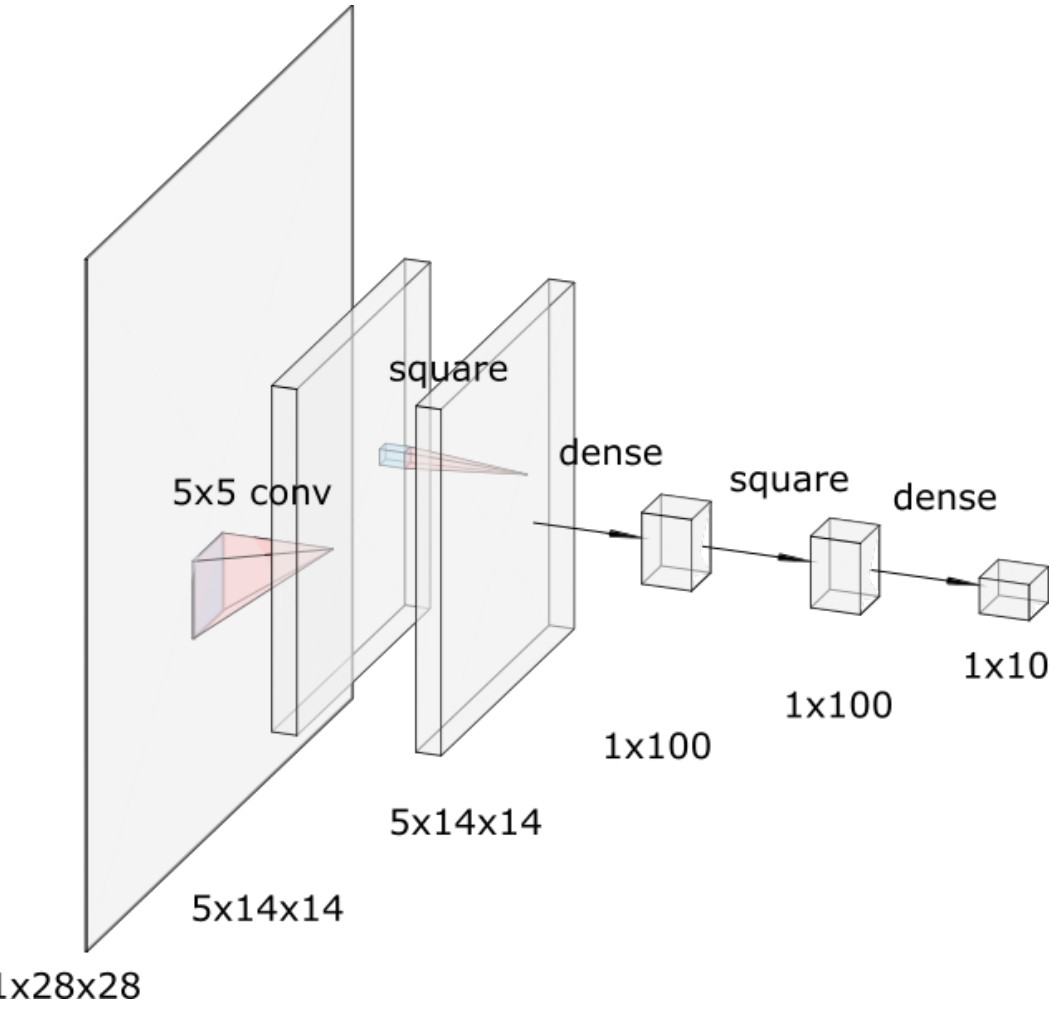

Figure 2: The structure of the network used for MNIST classification

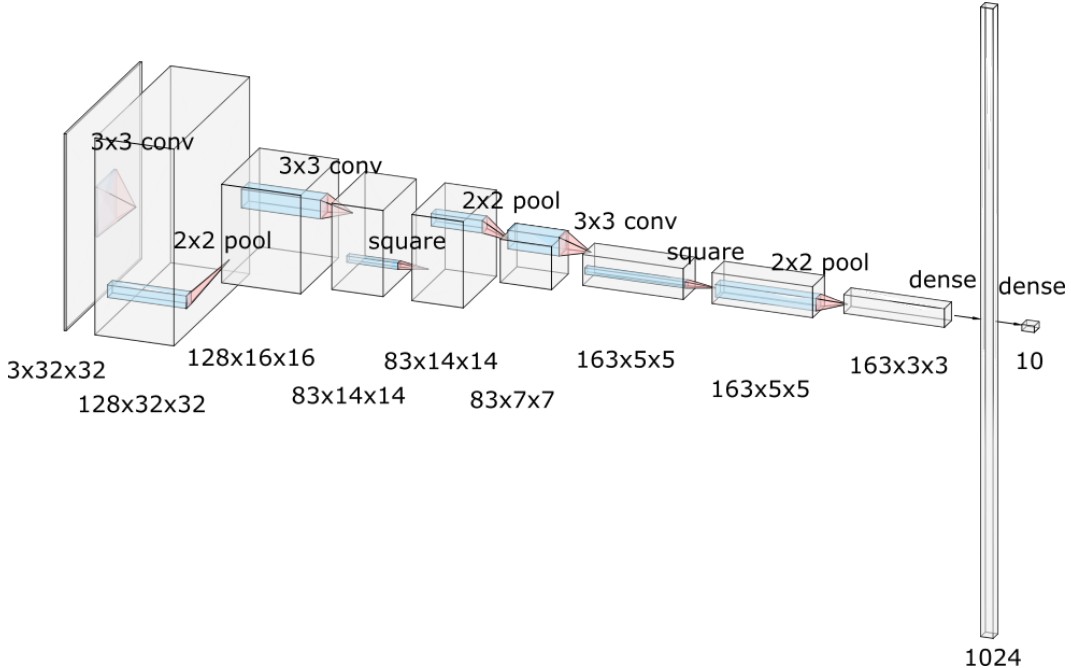

Figure 3: The structure of the network used for CIFAR classification.

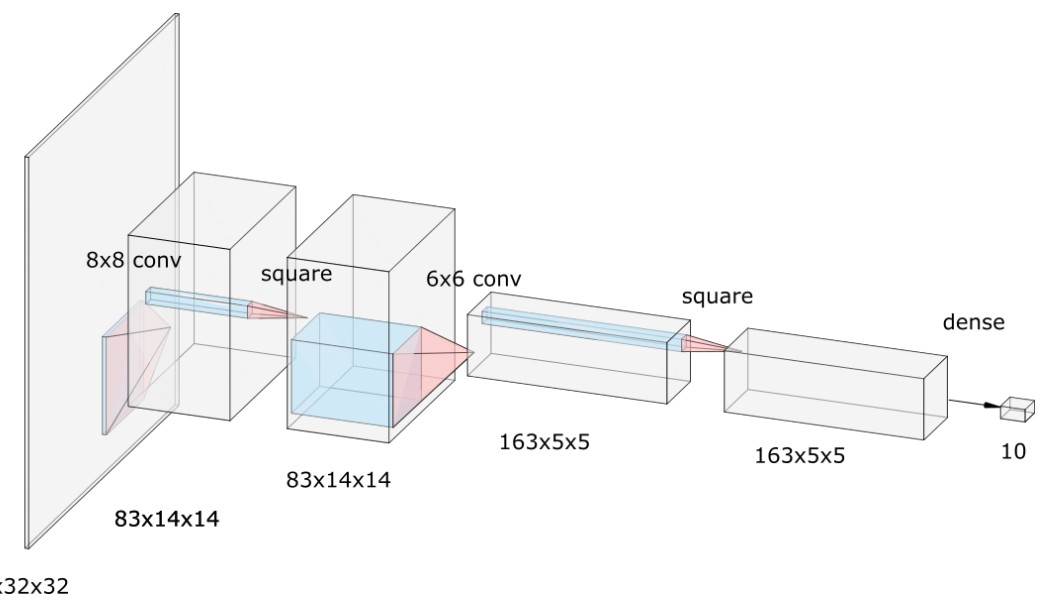

Figure 4: The structure of the network used for CIFAR classification after collapsing adjacent layers.

| Layer | Input size | Weights format | Output format | Description |
|---|---|---|---|---|
| Encryption | 784 | | dense | image is encrypted into a single message |
| $5 \times 5$ convolution layer | 784 | convolution(column-major) | convolution-interleave | mask input to create 25 messages |
| | $25 \times 169$ | | interleave | |
| | $5 \times 169$ | | interleave | combine 5 messages into one |
| square layer | 845 | | interleave | |
| | 845 | | stacked-intetleave | stack 16 copies |
| dense layer | $16 \times 845$ | row-major(stacked) | interleave | output in 7 messages |
| | $7 \times 16$ | | interleave | combine 7 messages into one |
| square layer | 100 | | interleave | |
| dense layer | 100 | row-major | sparse | |

Table 2: LoLa data representation changes. The table shows the different formats of the data during its evaluation with LoLa-Conv

| Layer | Input size | Weights format | Output format | Description |
|---|---|---|---|---|
| Preprocess | 784 | | convolution | |
| Encryption | $25 \times 169$ | | convolution | image is encrypted into 25 messages |
| $5 \times 5$ convolution layer | $25 \times 169$ | convolution(column-major) | dense | output is in 5 dense messages |
| | $5 \times 169$ | | dense | combine 5 messages into one |
| square layer | 845 | | dense | |
| | 845 | | stacked | stack 8 copies |
| dense layer | $8 \times 845$ | row-major(stacked) | interleave | output in 13 messages |
| | $13 \times 8$ | | interleave | combine 13 messages into one |
| square layer | 100 | | interleave | |
| dense layer | 100 | row-major | sparse | |

Table 3: LoLa-Conv data representation changes. The table shows the different formats of the data during its evaluation with LoLa

| Layer | Input size | Weights format | Output format | Description |
|:---:|:---:|:---:|:---:|:---:|
| Preprocess | $200 \times 300$ | | dense | apply convolution layers from Alex-Net |
| Encryption | 1000 | | dense | image is encrypted into 1 message |
| dense layer | 1000 | row-major | sparse | |

Table 4: LoLa-CalTech data representation changes. The table shows the different formats of the data during its evaluation with LoLa on the CalTech-101 dataset

