# OpenReview forum: "Low Latency Privacy Preserving Inference"
_ICLR.cc/2019/Conference_

### Official Review · AnonReviewer2 · 2018-10-27
**Potential to advance the area, but the paper should be improved**

**Rating:** 5
**Confidence:** 4

**Review:**

The paper proposes improvements on the area of neural network inference with homomorphically encrypted data, where a major drawback in current applications is the high computational cost.

The paper makes several minor contributions and, in my opinion, it fails to provide an unified message to the reader. The arguably independent contributions are:
1. the use of the faster SAEL 3.2.1 over CryptoNet — not really an innovation per se
2. flexible representation of message space — the main contribution
3. a much more complex network than CryptoNet for an experiment on CIFAR10 — minor contribution/application of 2
4. using pre-trained model for embedding images before encryption — minor contribution

I think the authors should refocus this work on point 2 and 3 only. 1 could simply be a note in the paper, as not a a real contribution and 4 may be even excluded as it goes toward a different direction.

I will mainly focus on the central contribution of the paper — the data representation — which in my opinion has the potential to progress this area further. Unfortunately, the current quality of presentation is suboptimal. The choices on architecture and intermediate representations of LoLa in Section 4 are hard to follow, and it is not clear to me when the authors are making a choice or satisfying a constraint. That is, for example when we aim to vs. we have to use one representation in place of another? Since this is the main contribution, I suggest the author to help the presentation with diagrams and formulae in this Section. Each component should be presented in modular fashion. In fact, this is what the authors did in Section 3. Yet, towards reuse of these concepts for machine learning applications, the authors could present their improvements as applied to *neural network layers*. E.g. responding to answers such as: what are the most efficient data representation in input and output if we want to compute a convolutional layer? It would be also interesting to break down cost requirements for each intermediate computation, to highlight bottlenecks and compare alternative choices.

Incidentally, LoLa-Conv improvements appear to be simply due to the use of a conv vs. dense layer as input.

Section 6 explains the use of a pre-trained AlexNet to embed data for learning a linear model (HE-aware) for Caltech 101. In the context of application of HE, the idea is novel to my knowledge, although it is a rather common practice in deep learning. The number reported in this section have no baseline to compare with and therefore it’s hard to evaluate their significance.

A missing reference is [A].

Minors
* section 4: “this saves a processing step which saves"
* typos in section 6: “a networks”, “in medical imagine"
* footnote 4: “convinient"

[A] TAPAS: Tricks to Accelerate (encrypted) Prediction As a Service.

---

> ### Author Response · Authors · 2018-11-22
> **Response to reviewer 2**
>
> Thank you for many valuable comments! We used your insights to improve the paper and we hope that we were able to address your concerns. The main concern you raise is about the contribution of the paper. The main contribution is a set of novel techniques that allow Encrypted Prediction as a Service (EPaaS) which are order of magnitude faster and significantly more scalable than previous state-of-the-art. Following your helpful comments, and the comments of other reviewers, we rewrote the abstract & intro and made small changes in the rest of the paper, to make the contribution of this work clearer.
>
>   We agree with your comment that using a faster library isn’t an innovation which is the reason we presented it as CryptoNets 2.3 and did not use the “brand name” LoLa which we use for other algorithms. Our motivation to introduce CryptoNets 2.3 is to allow the reader to distinguish between the improvement which is due to “better engineering” and the contribution that is due to our novel techniques. Our experiment on MNIST shows that the contribution of the better engineering techniques, that is CryptoNets 2.3, is significant (about order of magnitude speedup). However, the techniques we present add another order of magnitude speedup on top of that. The CIFAR experiment shows that these new techniques can handle harder tasks that CryptoNets & CryptoNets 2.3 cannot handle. Therefore, the CIFAR experiment is not a contribution by-itself, instead, it serves to demonstrate the significance of the contribution of the LoLa approach as it is not only faster but also more scalable.
>
> In your comments you wrote “Incidentally, LoLa-Conv improvements appear to be simply due to the use of a conv vs. dense layer as input.”. We would like to clarify that both LoLa and LoLa-Conv implement the same neural-network and therefore, in both cases they use a convolution layer and in both cases the convolution layer is implemented on the encrypted data. The difference between these solutions is that in LoLa the data is encrypted as a simple pixel-array while in LoLa-Conv the pixel order is rearranged before encryptions in a way that makes processing the image easier.
>
> The experiment on the CIFAR dataset demonstrates that the representation techniques we use in LoLa allow handling tasks that were too big for previous solutions. First and for most, we think that this is another demonstration of the significance of the contribution of this work. At the same time, operating on the CIFAR dataset is already much slower than on MNIST which suggests that even our approaches would not be able to handle the kind of nets used for tasks such as object-recognition in large images. Therefore, we propose a novel approach to handle deep networks that we demonstrate on the CalTech-101 dataset. This approach borrows the idea of using pre-trained networks that is typically used for transfer-learning. We applied this technique for the task of encrypted predictions. You commented that we did not present any benchmark for this task – this is true since to the best of our knowledge no one ever attempted to apply deep neural networks of similar depth with homomorphic encryptions as it was considered infeasible with current technology. Moreover, from previous results on simpler tasks such as CIFAR & MNIST, one would expect that predictions on CalTech-101 would take minutes or hours at best while we demonstrate a method that can make such predictions in 0.18 seconds.
>
> You commented that the choices of the different representation in LoLa are not clear. We added tables to the appendix which show the different steps and how the representations evolve during the computation. At the same time, we tried to keep the details in the paper to allow reproducing our results.
>
> A good comment was made about a missing reference to the “TAPAS: Tricks to Accelerate (encrypted) Prediction as a Service" paper. We now discuss this paper in the prior-art section. At the same time, we removed the comparison to the CryptoDL work since after attempting to reproduce the result and seeking the assistance of the authors it looks like the way they measured latency and accuracy makes it incomparable.
>
> To summarize, we would like to thank you again for your thoughtful and helpful comments. We think that some aspects of our presentation made it hard to identify the significant contribution of this work. We introduced changes to the presentation which hopefully address your concerns.

---

### Official Review · AnonReviewer3 · 2018-11-05
**Low latency version of CryptoNet**

**Rating:** 6
**Confidence:** 2

**Review:**

This paper proposes a low latency data representation / network architecture to operate on encrypted data. This work is based on CryptoNet, but uses a different data representation. How to apply machine learning on confidential data is important in many practical applications.

I recommend to give a brief review of HE before Section 3, to be more friendly.

---

> ### Author Response · Authors · 2018-11-22
> **Response to reviewer 3**
>
> Thank you for your comments. Your main concern was about the lack of enough background on HE. We added a subsection to the introduction that provides such an introduction. We think that this makes the paper easier to follow and we thank the reviewer for the helpful comment.

---

### Official Review · AnonReviewer4 · 2018-11-16
**Incremental improvements or poor motivation. Writing needs improvement.**

**Rating:** 5
**Confidence:** 3

**Review:**

This paper proposes several improvements to cryptonet proposed in Dowlin et al, 2016. The contributions include:
1. Better implementation to improve speed and throughput.
2. Modified architecture (LoLa) to reduce latency.
3. Using features from a deep network rather than raw input.
Contribution 1 is better engineering with modern libraries and more efficient parallelism. This has limited academic novelty. Contribution 2 seems interesting, but is poorly explained. What is the cause of this improvement? What are some guidelines of latency-bandwidth trade-off in homomorphic deep networks? Contribution 3 seems to eliminate the need for remote classification service itself. Users need classification services because they do not have the computation resources, or do not have enough data to train classifiers successfully. If the users need to generate good representations such that classification becomes linearly separable, then why don’t they just train the last linear layer themselves? Is there any computation or statistical reason to use any remote service?

The biggest problem with the paper is that it is hard to read and has several writing issues:
1. It is not self contained. I had to refer to Dowlin et al, 2016 to understand what the authors are referring to.
2. Notations are used but not defined. For example, I couldn’t find any definition several symbols in section 3.
3. The narration is too long and detailed: the authors report a laundry list of the things they did, details that should go into the appendix. It’s hard to find what the main contributions are.


------------ Response to rebuttals

The writing of the introduction has been greatly improved.

The authors suggested a practical scenario of using high level features. I am still somewhat skeptical. To do this the users need to map raw input to good representations. It seems that there are two ways this can work: users and service providers agree on publicly available representations, or the service provider is willing to share everything except the top layer. Both assumptions seem rather restrictive. Nonetheless even with practically useful scenarios, it is standard practice to use good features/representations whenever available, so not really an academic contribution. I am evaluating this paper only by the Lola contribution.

The presentation of LoLa can still be improved, but I see the main ideas. I think the paper could benefit from additional discussions and experiments. For example, when a practitioner wants to solve a new problem with some design need (e.g. accuracy, latency vs. bandwidth trade-off), what network modules can he choose and how to represent them? I think this type of general discussion can improve the significance and usefulness of the proposed approach.

---

> ### Author Response · Authors · 2018-11-22
> **Response to reviewer 4**
>
> Thank you for your thoughtful comments. Based on your comments, and the comments of other reviewers, we have updated the paper, especially the abstract and intro, to make it more self-contained and to make a clearer statement of the novelty of this work.
>
> To make it clear, we think that our work demonstrates significant contribution to privacy preserving predictions. This is achieved, for the most part, using new set of methods to represent matrices and vectors and moving between these representations. As a reference, we provided not only the performance of CryptoNets but also the performance of the same technique when applied using modern tools. We use the name CryptoNets 2.3 for this version to emphasize that this is a revision of CryptoNets. However, the new methods we propose, under the “brand-name” LoLa, are more than 11x faster than CryptoNets 2.3 and can handle larger networks as demonstrated, for example, on the CIFAR dataset.
>
> In your review, you commented about the idea of using “deep representations” as we demonstrate on the CalTech-101 challenge. You wrote “If the users need to generate good representations such that classification becomes linearly separable, then why don’t they just train the last linear layer themselves? Is there any computation or statistical reason to use any remote service?”. Training a linear model still requires a dataset that the user might not be able to collect. For example, imagine a dentist that would like to build a classifier over x-ray images to detect rare conditions such as tooth fusion. Since this disorder happens to about 0.5-2.5% of the patients, creating even a small sample of 1000 examples requires having access to 40,000-250,000 patients. Clearly, not every dentist has access to so many patients. This is where a service provider can step in and train a model by aggregating data from many dentists, a process which is time consuming and costly. Therefore, the service provider would rightfully treat the model as a valuable intellectual property and would not share the parameters of the model. Instead, the service provider may prefer to provide access to the model as a web-service in which case the mechanism we propose allows preserving the privacy users’ data (the xray images) while protecting the intellectual property of the service provider (the model).
>
> We understand that many readers in the ICLR community have little background in “applied homomorphic encryption”, at the same time we feel obliged to present reproducible results. To help with this tradeoff, we rewrote the abstract and intro and added Tables 2,3, and 4 that allow a quick look into the way we implement the different networks while skipping many of the technical details. We hope that these changes present a good tradeoff between reproducibility and clarity.

---

### Public Comment · (anonymous) · 2018-10-22
**The contribution**

What's your technical contribution compared with CryptoNet? It seems the only difference
comes from the used homomorphic library?

---

> ### Author Response · Authors · 2018-10-25
> **The main contribution is in using alternative representation methods that result in >10x performance gain**
>
>
> Replacing the homomorphic encryption library gives a speedup of about 8x, from 205 seconds to 24.8 seconds. However, the techniques presented in this work give an additional 11x speedup, down to 2.2 seconds per prediction for the MNIST dataset. The main difference between the approach we take here and CryptoNets is that CryptoNets encrypt every node in the network as a message while we encrypt entire layers. We show that this allows us to use HE’s ability to perform vector operations. We show that there are multiple ways to represent vectors in this scheme and to get low latency it is better to use a diverse set of representations.
>
> The difference between the approach we propose and CryptoNets is demonstrated again in section 5 when we try to apply secure predictions to the CIFAR dataset. Since the network used is more complicated, CryptoNets’ approach fails to execute since the amount of memory needed is in the order of 100’s of GBs, whereas our solution executes in 12 minutes (and uses only several GBs of RAM).
>
> In section 6 we push the idea of representations even further and show that if you use standard networks, such as AlexNet, to encrypt the image in a deep representation, you can obtain predictions in a fraction of a second. In contrast, if we were to apply the same network on the pixel level data as CryptoNets does, it will fail both because of memory requirements and lengthy computation.

---

### Public Comment · (anonymous) · 2018-10-22
**Missed a very relevant recent published paper**

Hi,
I do not see the mention of [1] at all in the paper. The paper was presented in the last ICML and I think should be a part of the literature review and comparisons.

[1] TAPAS: Tricks to Accelerate (encrypted) Prediction As a Service. http://proceedings.mlr.press/v80/sanyal18a.html

---

> ### Author Response · Authors · 2018-10-25
> **Thanks for the reference**
>
> Thank you for the reference, we will add it to the paper since it adds significant insights. The paper you pointed to argues that CryptoNets, and other similar solutions, leak information about the structure of the network the provider uses via the encryption parameters [1]. Therefore, they propose a new approach that uses a different encryption scheme that is capable of relatively fast bootstrapping but is limited to work with only binary values. To accommodate that, they use Binary Neural Networks (BNNs). After applying various techniques, they manage to perform a single prediction in 37 **hours** that is predicted to reduce to 2.41 **hours** if 16 servers are used in parallel to perform the prediction.
>
> In contrast, the approach we present here can make a prediction on a single machine in 2.2 **seconds**. Indeed, it leaks some information on the structure of the network, a point that we mention in the Conclusions section (Section 7). One thing to note, is that the TAPAS approach (the approach presented in the recent ICML paper) does leak information about the structure of the network too. For example, computation time varies with the network size, as can be seen in Table 2 in the reference paper that shows that the computation time is different for different tasks. Moreover, since the service allows an adversary to get predictions on its data-points, the adversary can use this to build a labeled dataset and train a model on it that will replicate the predictions the service provides.
>
> To conclude, we think that this is an important reference and we will include it in the paper. However, the problem they address is different and therefore it does not stand in direct comparison with the solutions we propose.
>
>
> [1] It is important to note that the information that leaks is only about the layout of the network and not about the data that was used for training or the data that was supplied for predictions.

---

> > ### Public Comment · (anonymous) · 2018-10-25
> > **Precise Summary!**
> >
> > I believe this is a very good summary of the paper. I would want to add that not leaking information through the encryption parameters also allows you to update the model continuously without informing the client. It's a small point but that might be very useful in the real world.

---

> > > ### Author Response · Authors · 2018-11-22
> > > **literature review updated**
> > >
> > > Good point, the revision of the paper we uploaded contains an updated literature review with reference to this work and its merits.

---

### Meta-Review · Area_Chair1 · 2018-12-13
**Incremental contribution**

**Confidence:** 3
**Recommendation:** Reject

**Metareview:**

The paper proposes improvements on the area of neural network inference with homomorphically encrypted data. Existing applications typically have high computational cost, and this paper provides some solutions to these problems. Some of the improvements are due to better "engineering" (the use of the faster SAEL 3.2.1 over CryptoNet). The idea of using pre-trained AlexNet features is new, but pretty standard practice. The presentation has been greatly improved in the updated version, however the paper could benefit from additional discussions and experiments. For example, when a practitioner wants to solve a new problem with some design need (e.g. accuracy, latency vs. bandwidth trade-off), what network modules should be used and how should they be represented? To summarize, the problem considered is important, however, as pointed out by the reviewers, both the empirical and the theoretical results appear to be incremental with respect to the existing literature.